# Clinical, Histopathologic and Genetic Features of Rhabdoid Meningiomas

**DOI:** 10.3390/ijms24021116

**Published:** 2023-01-06

**Authors:** Patricia Alejandra Garrido Ruiz, María González-Tablas, Alejandro Pasco Peña, María Victoria Zelaya Huerta, Javier Ortiz, Álvaro Otero, Luis Antonio Corchete, María Dolores Ludeña, María Cristina Caballero Martínez, Alicia Córdoba Iturriagagoitia, Inmaculada Catalina Fernández, Joaquín González-Carreró Fojón, Aurelio Hernández Laín, Alberto Orfao, María Dolores Tabernero

**Affiliations:** 1Neurosurgery Service of the University Hospital of Salamanca, Surgery Department, University of Salamancaca (USAL), Paseo de la Transición Española, 37007 Salamanca, Spain; 2Institute for Biomedical Research of Salamanca, IBSAL University Hospital of Salamanca, Paseo de San Vicente, 58-182, 10ªPlanta, 37007 Salamanca, Spain; 3Centre for Cancer Research (CIC-IBMCC; CSIC/USAL; IBSAL) and Department of Medicine, Campus Miguel de Unamuno, University of Salamanca, 37007 Salamanca, Spain; 4Biomedical Research Networking Centre on Cancer—CIBERONC (CB16/12/00400), Institute of Health Carlos III, C. Sinesio Delgado, 4, 28029 Madrid, Spain; 5Pathology Service of the University Hospital of Pamplona, Universidad Pública de Navarra, C. de Irunlarrea, 3, 31008 Navarra, Spain; 6Pathology Service of the University Hospital of Salamanca, Cell Biology and Pathology Department, Paseo de la Transición Española, 37007 Salamanca, Spain; 7Pathology Service of the Hospital Puerta del Mar, Av. Ana de Viya, 21, 11009 Cadiz, Spain; 8Pathology Service of the Hospital Álvaro Cunqueiro, Estrada de Clara Campoamor, 341, 36312 Vigo, Spain; 9Pathology Service of the University Hospital 12 Octubre, Universidad Complutense, Av. de Córdoba, s/n, 28041 Madrid, Spain

**Keywords:** rhabdoid meningioma, chromosome copy number alterations, diagnosis, prognosis, survival, histopathology

## Abstract

Rhabdoid meningiomas (RM) shows heterogeneous histological findings, and a wide variety of chromosomal copy number alterations (CNA) are associated with an unpredictable course of the disease. In this study, we analyzed a series of 305 RM samples from patients previously reported in the literature and 33 samples from 23 patients studied in our laboratory. Monosomy 22-involving the minimal but most common recurrent region loss of the 22q11.23 chromosomal region was the most observed chromosomal alteration, followed by losses of chromosomes 14, 1, 6, and 19, polysomies of chromosomes 17, 1q, and 20, and gains of 13q14.2, 10p13, and 21q21.2 chromosomal regions. Based on their CNA profile, RM could be classified into two genetic subgroups with distinct clinicopathologic features characterized by the presence of (1) chromosomal losses only and (2) combined losses and gains of several chromosomes. The latter displays a higher frequency of WHO grade 3 tumors and poorer clinical outcomes.

## 1. Introduction 

Meningiomas are the most common subtype of central nervous system (CNS) tumors which display heterogeneous histopathologic and genetic/molecular features that are associated with a usually benign clinical outcome. Thus, most meningiomas show World Health Organization (WHO) benign grade 1 histopathologic features, while WHO grade 2 and 3 meningiomas are less frequently observed [1,2]. Among all meningiomas, rhabdoid meningiomas (RM) are a rare tumor subtype which usually present with unique histopathologic characteristics. Thus, in the WHO 2021 classification of CNS tumors, RM are defined based on the presence of rhabdoid cells, i.e., “plump cells with an eccentric nuclei, open chromatin, macronucleoli, and prominent eosinophilic paranuclear inclusions, appearing either as discernible whorled fibrils or compact and waxy spheres”, and RM are classified as WHO grade 3 tumor [1,3]. However, the majority of RM cases usually show a combined rhabdoid cytology with variable percentages of rhabdoid cells that are associated with tissue areas which show histopathologic features that are characteristic of other histological variants of meningiomas (e.g., meningothelial, transitional, clear cell, or papillary) [4,5,6]. Among RM, rhabdoid cells usually represent >50% of the whole tumor cellularity and are the hallmark of this rare meningioma subtype. In turn, RM typically show a high mitotic rate (≥20 mitotic figures in 10 consecutive high-power fields, HPF), overt anaplasia, homozygous deletion of the *CDKN2A* and/or *CDKN2B* genes, and/or mutations of the promoter of the *TERT* gene [1,7]. Several authors have recently suggested that a rhabdoid cell morphology/histopathology might represent a phenotypic variant rather than a specific subtype of meningioma [8,9,10] since some RM cases that were first diagnosed as WHO grade 3 tumors [8,9,10,11,12,13] were reclassified after they were re-evaluated as grade 2 or even grade 1 meningiomas [8] with a rhabdoid cytomorphology in the absence of histological characteristics of anaplasia [12,13]. 

Likewise, variable and even controversial results have been reported about the clinical behaviors and outcomes of RM patients [5,9,14]. In this regard, it should be noted that the rhabdoid component might already be present at first diagnosis [15] or at recurrence [16] in men and women with a broad age range, including adults and children [8,9,10,17]. In addition, the number of rhabdoid cells might increase in some patients in subsequent recurrent tumors [5,14]. From a prognostic point of view, not every RM displays an aggressive clinical course. Thus, despite a tendency toward local recurrence [7,10,17,18,19], and because even distant metastasis in RM has been reported previously [5,10,20,21], the specific mechanisms leading to tumor relapse still remain to be elucidated. This is due at least in part to the inclusion in previously reported series of RM cases of patients whom could not always attain complete tumor resection and of cases with insufficiently long follow-up times required to reach definitive conclusions about patient outcomes. 

In recent decades, important advances have been achieved in the characterizations of genetic/molecular alterations of meningiomas [22,23,24,25]. Despite this, limited data exist regarding the specific genetic features of RM. Thus, preliminary studies have reported losses of chromosome 22 [26], BRCA1 associated protein 1 (*BAP1*) gene mutations and/or *BAP1* deletion [27,28], and a high expression of the matrix metallopeptidase 9 (*MMP9*) gene, which are characteristic findings of RM [29]. In addition, mutations of the *PBRM1* gene, which are predominantly found in papillary meningiomas, have been also reported in single RM cases [30]. Likewise, *BAP1* gene mutations and/or deletions, which are typically found in RM, have also been reported in papillary meningiomas in combination with alterations of the *PBRM1* gene [27,30]. Despite all these data, the presence and frequency of chromosome copy number alterations (can) other than losses of chromosome 22 still remain to be investigated in RM. 

In this study, we report on the histopathologic, clinical, and prognostic features of RM based on a large retrospective series of 305 cases of patients with 12 different series of RM [5,8,10,12,13,14,16,17,27,29,31,32], several RM case reports [15,33,34,35]) that were identified after an extensive review of previous reports, and a group of cases that were retrospectively collected from 13 centers in Spain. Based on the cases from Spain, we further investigated the genetic profile of RM as analyzed by whole genome copy number microarrays and its relationship with other features of the disease and patient outcomes for the first time.

## 2. Results and Discussion 

### 2.1. RM Cases Previously Reported in the Literature

A total of 233 tumors from patients with available histopathologic and/or clinical features consistent with RM were analyzed out of 305 collected cases (Table 1, Appendix A). These included a majority of adult cases (88%) and 12% of cases characterized by childhood RM (Appendix A), with median ages of 52 ± 16 years (range: 20–85 years) and 12 ± 4 years (range: 2–17 years), respectively. Overall, a predominance of women (56% vs. 44% men) was observed in adult tumors, but not in childhood tumors (48% vs. 52%, *p* = 0.441). According to the WHO 2021 classification, 32% of the cases were WHO grade 3 meningiomas, while 41% were classified as WHO grade 2 tumors, and 27% were classified as WHO grade 1 tumors. A mixed histological pattern was found in a great majority of the tumors (89%). In many tumors, a low mitotic index with a few scattered mitotic cells was found as 109 of the 163 tumors assessed (67%) displayed <4 mitoses/10 HPF and only 8% had ≥20 mitoses/10 HPF (this analysis was based on the 163 cases with available information). Most patients (105/159 cases, 66%) underwent gross (total) tumor resection, while a partial tumor resection had been performed in 54/159 cases (34%). Adjuvant treatment was administered for 104/196 (53%) patients, and radiotherapy was the most frequent adjuvant therapy (100/196 cases, 51%). After a variable follow-up period (including a median of 28 months and a range 2–204 months), 43/102 patients had undergone tumor recurrence. These patients included 11/19 children (58%) and 32/83 adults (39%) (*p* = 0.124). Most patients (97/133, 73%) were alive at the moment the data had been collected and the results were reported (Appendix A).

Altogether, these data show that meningiomas that carry rhabdoid features consist of a highly heterogeneous group of tumors that involve both children and adults with a slight female predominance and highly variable histopathologic features that are associated with a predominance of WHO grade 2 and grade 3 tumors and an increased rate of tumor recurrence and death compared with other, more prevalent meningioma types [36,37]. For a more robust analysis of RM, particularly regarding disease outcome, we focused our analysis on a more homogeneous group of tumors that consisted of adults studied at diagnosis who had undergone complete tumor surgical resection, which included a total of 71 patients (Figure 1). Among these patients, a slight (*p* > 0.05) predominance of women (55%) with a similar distribution per age interval (peak at 51–60 years) for WHO grade 1 and WHO grade 2 tumors (but not WHO grade 3 tumors, age peak at 41–50 years, *p* > 0.05) was observed along with a predominance (52%) of men (Figure 1a,b). This was associated with a low mitotic rate of less than 4 mitoses/10 HPF in more than half (53%) of the patients and a predominance of low-proliferative tumors among WHO grade 1 (100%) and grade 2 (43%) vs. grade 3 RM (15%). As might have been expected, progressively higher frequencies of cases showing a greater mitotic rate (≥20 mitoses/10 HPF, *p* = 0.001) (Figure 1c) together with a higher proliferative index (>20% Ki-67^+^ cells) were observed in WHO grade 2 (9%) and grade 3 (48%) RM, but not in grade 1 RM (Figure 1d). This also translated to a tendency toward an increased frequency of cases undergoing tumor recurrence (12% vs. 31% and 35%) and deaths (13% vs. 22% and 36%) in WHO grade 1 vs. grade 2 and grade 3 RM (*p* = 0.247), respectively (Figure 1e,f). After a median follow up of 43 months, a median recurrence-free survival (RFS) of 44 months (range: 3–171 months) was observed for a subset of patients (adults who underwent gross total resection who presented with rhaboid content at diagnosis and had a follow up >3 months) with slightly different 5-year RFS rates of 88%, 43%, and 23% for WHO grade 1, grade 2, and grade 3 RM, respectively (*p* = 0.06). When grade 2 and grade 3 RM were compared with grade 1 tumors (Figure 1g), they had statistically significant differences, and they did not for 5-year overall survival (OS) rates of 100%, 88%, and 68% for grade 1, 2, and 3 RM (*p* = 0.106) (Figure 1h). 

### 2.2. Clinical, Histopathologic and Genetic Features of Our Retrospective Cohort of RM Patients

From the 33 RM specimens analyzed in our laboratory (from a total of 23 meningioma patients), 25 tumors were classified as WHO grade 3, 5 tumors were grade 2, and 3 tumors were grade 1 meningiomas, and the median age was 65 years (range: 34–79 years) (Appendix A). These 25/33 WHO grade 3 specimens corresponded to a total of 15/23 patients who met WHO grade 3 criteria for RM (diagnostic and recurrence samples), including 10 men (67%) and 5 women (33%) (Table 2). A rhabdoid histology was the most prominent component (>50%) in 8 tumors (53%), which coexisted together with other histological meningioma subtypes, such as meningothelial and papillary components, in 40% and 33% of RM, respectively (Table 2). Two RM specimens showed a fibrous component, and another one had an angiomatous component. Low mitotic activity was observed in 11/15 tumors, while 4 tumors had >4 mitoses, and none presented with ≥20 mitoses. The percentage of Ki-67^+^ cells (MIB-1 staining) was higher than 30% in one tumor, and around half of the tumors (53%) had a Ki-67 labeling index (LI) between 10–15%. Immunohistochemistry (IHC) showed positive staining for vimentin (100%) and EMA (94%) in virtually all of the RM cases. Local recurrences developed in 40% of the patients (two patients presented one relapse, another patient had two, and another three patients had three relapses) and occurred within 3 to 9 years from the first surgery. After a median follow-up of 9 years, 53% of the patients remained alive, while four patients died due to the meningioma (after follow up times of 2.5, 11, 12, and 13 years, respectively), and another two RM died because of causes not related to the tumor. The current status (alive vs. dead) was unknown for one patient (Table 2).

Overall genetic data on whole genome CNA was obtained in a total of 29 specimens, including 2 WHO grade 1 meningiomas, 3 WHO grade 2 tumors, and 24 grade 3 RM, and the 24 grade 3 specimens corresponded to tumors from 15 patients. Of note, CNA that involved between 2 and 19 different chromosomes were identified in every altered tumor, and there was a median of 10 altered chromosomes per tumor (Figure 2a). The most common genetic aberration consisted of a loss of chromosome 22 (96%), which corresponded to the loss of the entire chromosome 22 in 76% of cases (19/24 tumors), while interstitial deletions involving this same chromosome (20%) were observed in 5/24 tumors (Figure 2b). Other frequently observed chromosomal losses involved chromosomes 14 (16/24 tumors, 64%), 1p and 6q/6 (15/24 tumors, 60% each), chromosome 19/19q (14/24 tumors, 56%), chromosome X (6/9, 67% of the female patients), chromosome Y (lost in 8/15 or 53% of the men), and chromosome 18 (11/24 tumors, 44%). In turn, gains were less frequently observed. They involved regions of chromosome 17/17q (10/12 tumors, 40%), 1q (9/24 tumors, 36%) and 20 (7/24 cases, 28%) (Appendix A). Besides the above mentioned altered, large chromosomal regions, additional yet minimal common recurrent regions (MCR) were also found to be altered, including losses at chromosome 22q11.23, 13q14.2, and 10p13, together with gains at chromosome 21q21.1 (Table 3). Based on their CNA profile, RM could be classified into two well-defined genetic profiles (Figure 2c) consisting of (1) RM tumors presenting with multiple chromosomal losses (8/24 tumors, 33%) with a median of 4 (range: 2–6) altered chromosomes in the absence of chromosome gains and (2) RM in which chromosomal losses and gains that involved several chromosomal regions coexisted (with a median of 10 affected chromosomes; range: 3–19 chromosomes) (Figure 2c). 

Several reports indicated that RM are usually WHO grade 3 meningiomas associated with an adverse prognosis [20,21,34,38,39], although controversial results exist in the literature in this regard [8,11,19]. Such discrepancies might be due, at least in part, to the underlying genetic heterogeneity of this rare subtype of meningiomas. In the WHO 2021 classification of CNS tumors, RM are defined as high grade (WHO grade 3) tumors which consist of widespread rhabdoid tumor cells in association with anaplastic features [8,11]. However, other meningiomas might show rhabdoid features in association with different rhabdoid cell contents in the absence of other histological characteristics of malignancy, which does not support a WHO grade 3 diagnosis in every RM [8]. In this regard, Vaubel et al. have proposed the use of different cut-off percentages of rhabdoid cells to subclassify RM, which confirmed that some RM actually did not display features that were consistent with WHO grade 3 meningioma (e.g., anaplastic) and would be classified as WHO grade 2 and even WHO grade 1 [40]. These results are consistent with our findings and those of other authors [41] which also confirmed that RM may show a wide range of histopathological features in which the rhabdoid tumor areas coexist with other meningioma tissue subtypes from which a papillary architecture is most frequently observed, which suggests a close association between both histological subtypes of meningiomas [4,5,40,41,42,43]. Considering this histopathological heterogeneity and the lack of specific immunohistochemical markers (e.g., the vast majority of meningiomas stain for EMA and vimentin) [5,8,10,14], controversial results might be due to the inclusion of misclassified tumors under this diagnostic subtype. In turn, other potentially useful criteria, such as a high mitotic rate and Ki-67 proliferative index, do not systematically correlate with the presence and representativeness of the rhabdoid tumor cell component [5,8]. Thus, RM with a low mitotic index of <4% mitoses/10 HPF have been reported in some studies together with highly proliferative tumors (with ≥20 mitoses/10 HPF and >20% Ki-67^+^ cells) [10,12,13,14] with a more aggressive clinical course and higher death rates which are similar to those of anaplastic meningiomas [44,45]. This has made it difficult to establish diagnostic cut-off values for the above parameters, including a more objective diagnostic classification of this subtype of meningiomas. In this study, we confirm these results based on the observation of a highly variable proportion of Ki-67^+^ cells among our cases, which highlights the need for more objective criteria for more accurate and reproducible diagnoses and classifications of RM in addition to the demonstration of the characteristic rhabdoid cytomorphological features [5,8,21,34,46]. 

From the prognostic point of view, previous studies suggest more aggressive behavior from RM that is associated with a high rate of local recurrence and even distant metastases [47,48,49,50,51], which is confirmed in our cohort. This is in line with previous observations which show that 40% of patients develop tumor recurrences [5,8,19,20,21,52,53], which is a relatively large fraction of our RM developed tumor recurrence in association with an increased rate of deaths. Despite this, it should be noted that the degree of surgical resection of the tumor might play a role in determining the variable frequency of relapses reported among RM cases. In line with this hypothesis, evidence indicates that a major prognostic factor for tumor recurrence is the extent of tumor resection, which depends on (and is influenced) by the tumor size, location, and its proximity to vital intracranial structures and vessels, in addition to the skill of the surgeon [5,8,10]. 

To the best of our knowledge, this is the first report in which detailed analyses of a relatively large number of RM was investigated for the presence of CNA throughout the whole genome. Overall, our results revealed the existence of two major CNA profiles for the 24 human chromosomes associated with (1) one or more chromosomal losses in the absence of chromosomal gains and (2) more complex genetic profiles, including multiple losses and gains of different chromosomal regions. A finding that is noteworthy is that monosomy 22/22q deletions were the most frequently observed chromosomal alteration, as also reported in other meningioma types [54,55]. Other frequently observed chromosomal losses included CNA involving extensive regions of chromosomes 1p, 6q, 14q, and 19p, together with gains of chromosomes 17, 1q, and 20. Additional CNA involving loss of MCR located at 22q11.23 and gains of the three minor common regions located at 13q14.2, 10p13, and 21q21.2 that might contain candidate oncogenes which are potentially involved in the pathogenesis of RM were described here for the first time. From a clinical point of view, the coexistence of CNA involving multiple (i.e., >7) chromosomal regions found to be lost was altered at significantly lower frequencies in meningioma subtypes other than RM [55,56,57] and was associated with a poorer prognosis and a high frequency of tumor recurrence and deaths when compared with RM displaying the loss of a more limited number of chromosomal regions. Interestingly, other genetic changes which have been previously reported among RM, such as the presence of homozygous deletions of *CDKN2A* in chromosome 9p [23,40] in association with WHO grade 3 tumors and unfavorable outcome [25,40], were rarely found to be deleted among our RM cases. Altogether, these results support the existence of unique but heterogeneous genetic profiles in RM which might help explain the heterogeneous histopathological features and the variable clinical behavior of this rare subtype of meningiomas, and thereby contribute to a better subclassification of RM.

## 3. Materials and Methods

### 3.1. RM Tumor Samples

First, we performed a deep review of RM cases previously reported in the literature between 1998 and 2021 [58,59,60,61,62,63,64,65,66,67,68,69,70,71,72,73,74,75,76,77,78,79,80,81,82,83,84,85,86,87,88,89,90,91,92,93,94]. A total of 305 RM cases were identified, including adults and children or adolescents under the age of 18 years. These RM cases had been included in 12 different series of ≥4 tumors and 63 reports of 1 to 3 cases (Appendix A). Out of the whole series, 233/305 samples had associated clinical and/or histological data. From them, 71/233 adult RM patients had been studied at diagnosis and underwent gross (total) tumor resection with available outcome data. 

In parallel, a retrospective series of 33 RM samples studied at diagnosis or at relapse from 23 patients who had been diagnosed at 13 different hospitals in Spain, including 24 tumors from 15 WHO grade 3 RM patients, were studied. Formalin-fixed and paraffin-embedded (FFPE) tumor tissues from the later cohort were obtained after surgery according to standard local protocols that varied and prevented us from obtaining reproducible data with some techniques/markers (i.e., staining for epigenetic markers like H3K27me3). Tissue examination and histopathological diagnosis were done locally at the Pathology Services of Hospital Álvaro Cunqueiro, the University Hospital Araba, the Hospital of Bellvitge, the General University Hospital of Ciudad Real, the University Hospital of Cruces, the Hospital Puerta del Mar, the Complejo Hospitalario de León, the University Hospital of Pamplona, the University Hospital 12 Octubre, the University Hospital of Salamanca, and the University Hospital Vall d’Hebron, or they were provided by the biobank of the Complejo Hospitalario Universitario Albacete, the biobank del Principado de Asturias, the biobank IIS Galicia Sur, and the biobank of Santiago (Spain). In addition, FFPE hematoxylin and eosin (H&E)-stained tissue sections from all 33 meningiomas were cut and centrally reviewed by two neuropathologists who evaluated the presence and number of rhabdoid cells, among other tumor features. For each tumor, the percentage of rhabdoid cells was semi-quantitatively estimated (<20%, 20–50%, >50%) together with other histological features, such as the mitotic rate per 10 HPF and the tumor grade as per the WHO 2021 criteria. Immunohistochemical staining for the epithelial membrane antigen (EMA), cytoplasmatic vimentin, and nuclear Ki-67 LI (detected using the MIB1 monoclonal antibody) were carried out in FFPE tissues following the manufacturers’ instructions. The percentage of Ki-67 immunopositive cells was used as an indicator of tumor cell proliferation. In addition, data on patient age, sex, treatment (including both the extent of surgical resection and the use of postoperative radiotherapy), and follow up data, including patient survival data at the moment of closing this study, were collected. For the purpose of this study, tumors that showed regrowth after the first complete tumor resection were classified as recurrent RM, which included a total of six patients who experienced between one and three recurrences. 

### 3.2. Fluorescence In Situ Hybridization (FISH) and CNA Array Studies

Tissue sections (3 µm) were cut from FFPE tumor samples and genomic DNA was isolated using the QIAamp DNA FFPE Tissue Kit (Qiagen, Valencia, CA, USA) according to the manufacturer’s instructions. The concentration of double-stranded genomic DNA was quantified using the Qubit^®^ dsDNA Assay (Invitrogen-Thermo Fisher Scientific, Waltham, CA, USA). A fraction of the extracted DNA from each RM sample (40–80 ng/tumor) was hybridized using the Affymetrix OncoScan array for genomic copy number assessments (Affymetrix-Thermo Fisher Scientific, Waltham, CA, USA). A total of 29 arrays were analyzed, and 24 corresponded to WHO grade 3 RM, three to WHO grade 2 meningiomas, and two to WHO grade 1 meningiomas. Four tumoral DNA samples were discarded due to poor gDNA quality. Briefly, the OncoScan FFPE copy number platform assay based on the Molecular Inversion Probe technology for small amounts of DNA from FFPE samples was used. A GeneAmp PCR system 9700 Thermal Cycler (Life Technologies, Carlsbad, CA, USA) was used for probe annealing up to the DNA denaturation stage. Digested DNA was hybridized on an OncoScan array and incubated at 49 °C in the Genechip Hybridization Oven 640 for 17 h at 60 rpm. OncoScan arrays were then washed in a GeneChips Fluidics Station 450 (Affymetrix) using the OncoScan stain and wash reagents according to the manufacturer’s instructions (Affymetrix). Finally, the microarrays were scanned on a GeneChip scanner 3000 (Affymetrix). Data QC was performed with OncoScan Console software per the recommendations of the manufacturer (Affymetrix). CNA were determined using normalized data via the Nexus Express for OncoScan software (version 3.1, Affymetrix). In addition, the OSCHP-TuScan data format and tool of the Chas Console software (Affymetrix) was used to identity CNA and calculate the percentage of altered cells and overall ploidy, including the percentage of cells that displayed a loss of heterozygosity in each sample. The Genome Reference Consortium Human Build 38 (GRCh38) was used to define the probe location. Gains were determined if the log2 ratio signal value obtained was >0.5, whereas losses were defined at the <−0.5 cut-off value. Significantly different regions of CNA were merged into a single genome interval whenever they were within a 0.5 Mb distance; subsequently, they were filtered to exclude small regions (<0.5 Mb) if another similarly altered region was not present within the nearest 10 Mb sequences. The lowest *p*-value (from a segment of at least 50 kb in length) within the merged regions was used to annotate the regions using Chromosome Analysis Suite (ChAS) (Thermo Fisher Scientific). Weighted log2 ratios were also obtained for each array probe using the ChAS console (v.4.2) (Affymetrix). These data were winsorized and segmented by the pcf (piecewise constant fragments) algorithm from the copynumber package (v.1.30.0) [95] in R (v.4.2.1, R Development Core Team, 2022 https://www.r-project.org, accessed on June 2022). The minimal common regions and the recurrent broad alterations were calculated by GISTIC (v.6.15.28) [96] across all samples at a confidence level of 0.90 and a q-value threshold of 0.05. FISH confirmatory analyses were performed on FFPE tumor sections to identify numerical alterations of chromosomes X and Y in double staining using chromosome specific probe-Vysis CEP X (Spectrum Aqua) and Vysis CEP Y (Spectrum Green), respectively (Abbott, Chicago, IL, USA), as previously described in detail [56].

### 3.3. Statistical Methods

SPSS software was used for all statistical analyses (SPSS 25.0, IBM SPSS, Armonk, NY, USA). The Chi-square test and the Mann–Whitney U test were used to compare different groups of patients for categorical and continuous variables, respectively.

## 4. Conclusions

In summary, the present study shows the presence of a rhabdoid meningioma cell component at variable proportions and in combination with components of other distinct histopathologic meningioma subtypes across all WHO tumor grades, which makes cytomorphological diagnosis of RM challenging in the absence of additional criteria. From a genetic point of view, RM showed monosomy 22/22q deletions in virtually all RM that is usually associated with deletions of chromosomes 1p, 6q, 14, and 19p alone or in combination with gains of chromosomes 17, 1q, and 20 and more complex karyotypes. Minor common regions of CNA were also identified in RM for the first time in this study, which involved 22q11.23 losses and gains at the 13q14.2, 10p13, and 21q21.2 chromosomal regions. The two different CNA profiles found among RM were associated with different distributions of WHO tumor grades as well as a distinct patient outcome, which highlights their potential utility for a more robust clinical subclassification of RM.

## Figures and Tables

**Figure 1 ijms-24-01116-f001:**
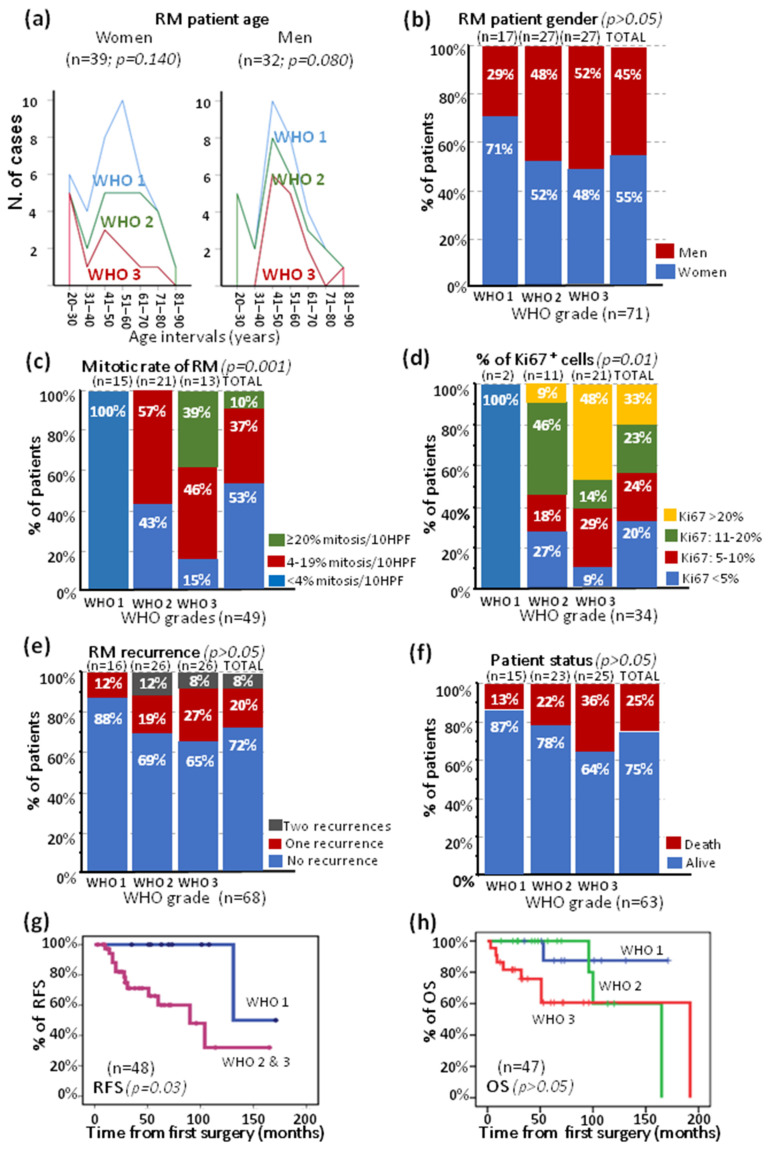
Clinical and histopathologic features of adult RM patients reported in the literature who had been studied at diagnosis after undergoing gross (total) tumor resection, distributed according to WHO tumor grade (n = 71). (**a**) Patient distribution per age and (**b**) per sex. The most relevant histopathologic features are included, such as (**c**) the mitotic rate and (**d**) the proliferative index (the percentage of Ki-67 positive cells). In panel (**e**), the distribution of recurrent vs. non recurrent tumors is shown, while in panel (**f**), the frequency of patients who were alive (vs. death) at the moment the case was reported is shown. (**g**,**h**) represent recurrence-free survival (RFS) and overall survival (OS) observed in a subset of adult RM patients who underwent gross total resection with a follow up longer than 3 months which are shown according to WHO tumor grades.

**Figure 2 ijms-24-01116-f002:**
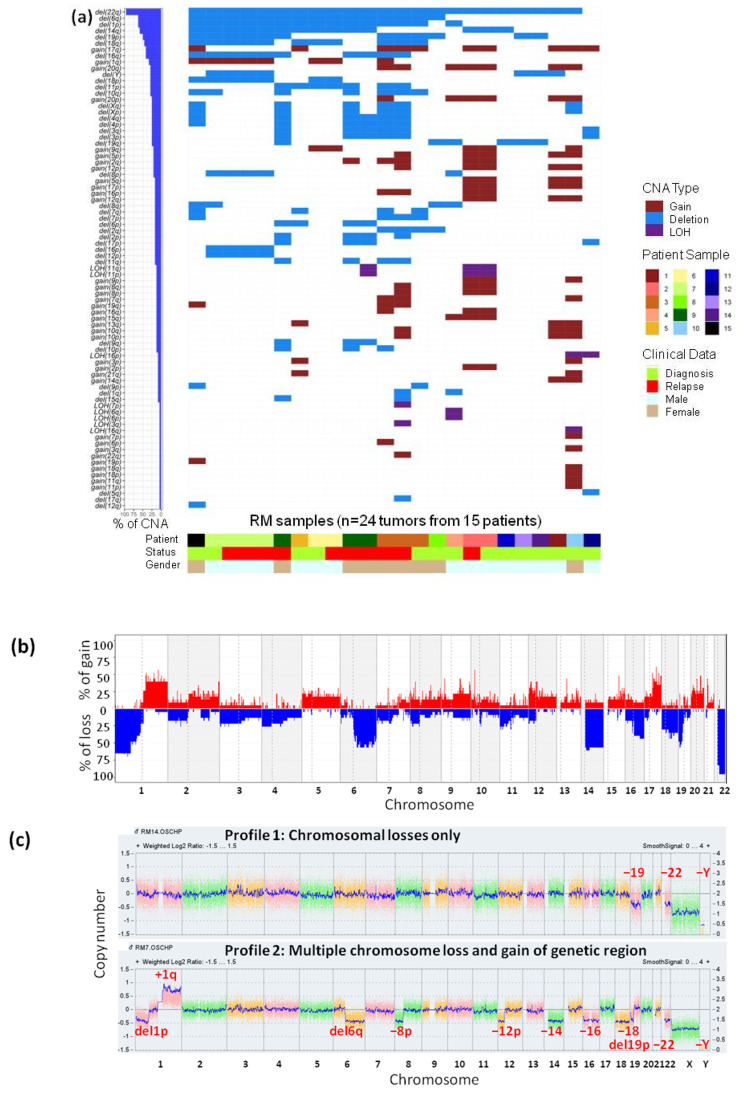
Whole genome plots illustrating chromosomal alterations detected in adult RM. (**a**) Summary plot of the complex and heterogeneous chromosomal alterations identified in WHO grade 3 RM tumors. Each column represents a tumor sample, and each row represents a different chromosomal region. The left barplot shows the frequency that the corresponding chromosome region was altered. (**b**) The frequency (%) of each CNA is represented by the amplitude found for each chromosomal region from chromosomes 1 to 22 (from left to right). Gains are represented by upward bars, while losses are represented by downward bars. (**c**) Illustrative examples of tumors presenting different genetic profiles that were defined according to the presence of chromosome losses only (profile 1) or combined (multiple) chromosomal losses and gains (profile 2).

**Table 1 ijms-24-01116-t001:** Rhabdoid meningioma patients previously reported in the literature between 1998 to 2021 with available (partial or complete) clinical data (n = 233).

Distribution per Age Group	Type of Study		Total
	Patient Series (n = 12)	Case Reports (n = 68)	
Adults	158 (95%)	46 (69%)	204 (75%)
Children	8 (5%)	21 (31%)	29 (10%)
Total	166 (71%)	67 (29%)	233 (100%)

**Table 2 ijms-24-01116-t002:** Clinicopathological findings and patient outcome of grade 3 RM studied in our laboratory after a long follow up (n = 15 patients).

Clinical and Histopathological Features		Percentage (N of Cases)
Age * (≤60 years or >60 years)		47% (7)/53% (8)
Gender (M/F)		67% (10)/33% (5)
Histology		
	Rhabdoid component (%)	
	>50%	53% (8)
	20–50%	47% (7)
	Other histologic components	
	Meningothelial	40% (6)
	Papillary	33% (5)
	Fibrous	13% (2)
	Angiomatous	7% (1)
N of mitotic figures		
	None detected	73% (11)
	>4 mitoses	27% (4)
	≥20 mitoses	0%
IHC markers		
	Ki-67^+^ cells	
	≤4%	20% (3)
	5–9%	20% (3)
	10–15%	53% (8)
	30%	7% (1)
	Vimentin ^+^	100% (15)
	EMA ^+^	93% (14)
	All cells ^+^	47% (7)
	Partial positivity	47% (7)
	Negative	7% (1)
Tumor recurrence		
	No	60% (9)
	Yes (1–3 recurrences)	40% (6)
Status		
	Alive ^#^	53% (8)
	Death due to RM	27% (4)
	Death due to other causes	13% (2)
	Unknown	7% (1)

* Median age was 65 years (range 34–79 years); M: male; F: female; ^#^: follow up of 4 patients (9 patients remained alive <4 years after diagnosis, while it was >8 years for the other five), +: positive IHC staining.

**Table 3 ijms-24-01116-t003:** Minimal common recurrent (MCR) regions found to be altered in RM.

Type of CNA	Chromosome	Location		Significance *
		Chr Band	Region Limits	
Gain	13	13q14.2	48,973,098–48,987,407	4.85 × 10^−14^
Gain	10	10p13	17,038,139–17,285,288	6.20 × 10^−12^
Gain	21	21q21.1	17,022,210–17,278,961	4.11 × 10^−7^
Loss	22	22q11.23	24,336,171–24,378,047	9.39 × 10^−7^
Gain	17	17q22	55,532,689–55,656,857	1.64 × 10^−6^
Gain	13	13q33.1	102,170,122–102,372,960	1.10 × 10^−5^
Gain	15	15q23	71,696,542–72,057,577	1.10 × 10^−5^
Gain	7	7p11.2	55,121,109–55,200,624	4.17 × 10^−5^
Gain	9	9p22.3	15,732,421–16,428,624	7.48 × 10^−5^

CNA: Copy number alteration; Chr: chromosomal; *: q values.

## Data Availability

The data presented in this study are available on request from the corresponding author.

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
