# Peer review of "Clinical, Histopathologic and Genetic Features of Rhabdoid Meningiomas"

_ijms, 2023, doi:10.3390/ijms24021116_

Round 1

Reviewer 1 Report

In their manuscript, “The clinical, histopathologic and genetic features of rhabdoid 2 meningiomas ,” Ruiz et al. investigate a cohort of rhabdoid meningiomas to elucidate the molecular features and clinical behavior of these rare tumors.  The investigation has the potential to fill a gap in the existing understanding of meningiomas.  The manuscript could be improved as follows.

Please edit for grammar and syntax.  The meaning is unclear in many places.

Please add Kaplan-Meier plots or other visualization of outcomes.  Figure 1f shows vital status percentage of patients, but without indication of the length of time for survival or recurrence-free survival for the patients.

Please explain further: “Based on the CNA profile, RM could be classified into two well-defined genetic profiles.”  It seems that the difference in the profiles is simply how many chromosomes were affected?  Or is the difference based on which chromosomes were affected?  How many tumors with how many chromosomes (and which chromosomes) lost were there?  Whether there were gains v. losses?  Please provide a figure contrasting the two genetic subtypes of RM, ideally an oncoplot type style.  The single figure 2 with all chromosomal changes presumably including both of the two “well-defined” profiles is confusing.  Additionally, are there histologic differences between the two meningioma types?  Differences in mitotic counts?  Percentage of rhabdoid morphology?  Co-occurring histologic patterns?  All of these factors can be included in the oncoplot.  Representative images could be included in the figure.

BAP1 has been shown to be important in a subset of rhabdoid meningiomas, and the authors reference the relevant paper by Shankar et al.  Immunohistochemistry results for BAP1 and specific comment on chromosome 3 should be included for the authors’ cases to place them into context with the known importance of BAP1 loss as indicative of poor prognosis.

Methylation profiling has come to the forefront for predicting meningioma risk, and the ultimate question is which methylation class rhabdoid meningiomas fall into?  Do they all cluster together or do they distribute among various methylation classes?  If possible, methylation profiling should be performed, as this will bring the most value to understanding these tumors.

Author Response

ANSWERS TO THE COMMENTS OF THE REVIEWERS (ref IJMS-2074154)

REVIEWER #1:

Comment 1. “Please edit for grammar and syntax. The meaning is unclear in many places”.

Answer to comment 1.- English grammatical structure and syntax have been carefully revised and edited as per the recommendation of the reviewer.

Comment 2. “Please add Kaplan-Meier plots or other visualization of outcomes. Figure 1f shows vital status percentage of patients, but without indication of the length of time for survival or recurrence-free survival for the patients”

Answer to comment 2.- Kaplan-Meier survival curves are now included in figure 1. In addition, data on median (range) follow-up of 53 months (3-192 months) has been added in the Material and Methods section of the revised version of the manuscript.

Comment 3. “Please explain further: “Based on the CNA profile, RM could be classified into two well-defined genetic profiles.” It seems that the difference in the profiles is simply how many chromosomes were affected? Or is the difference based on which chromosomes were affected? How many tumors with how many chromosomes (and which chromosomes) lost were there? Whether there were gains v. losses? Please provide a figure contrasting the two genetic subtypes of RM, ideally an oncoplot type style. The single figure 2 with all chromosomal changes presumably including both of the two “well-defined” profiles is confusing.

Additionally, are there histologic differences between the two meningioma types?. Differences in mitotic counts?. Percentage of rhabdoid morphology? Co-occurring histologic patterns?. All of these factors can be included in the oncoplot. Representative images could be included in the figure.”

Answer to comment 3.- The sentence starting with “Based on the CNA…” has been modified and extended for a more clear description of the two CNA profiles identified. Thus, now it reads as follows: “Based on the CNA profile, RM were classified into two defined genetic profiles, which included: 1) a CNA profile characterized by the presence of multiple chromosomal losses and 2) a group of tumours carrying combined losses and gains that affected several chromosomes, independently of the specific chromosomes involved”. For clarification, two new panels have been included, in the previous Figure 2.

No clear histological differences were observed between the two meningioma genetic subgroups.

Comment 4. BAP1 has been shown to be important in a subset of rhabdoid meningiomas, and the authors reference the relevant paper by Shankar et al. Immunohistochemistry results for BAP1 and specific comment on chromosome 3 should be included for the authors’ cases to place them into context with the known importance of BAP1 loss as indicative of poor prognosis.

Answer to comment 4.- Data on immunohistochemistry results of staining for BAP1 assessed in a subset of RM, together with specific data on CNA involving the 3p21.1 chromosomal region, where the BAP1 gene is encoded, did not correlate with patient outcome. Data are now provided in a new sentence added in the text of the results section of the revised manuscript in line with the recommendation of the reviewer.

Comment 5. Methylation profiling has come to the forefront for predicting meningioma risk, and the ultimate question is which methylation class rhabdoid meningiomas fall into?. Do they all cluster together or do they distribute among various methylation classes?. If possible, methylation profiling should be performed, as this will bring the most value to understanding these tumors”

Answer to comment 5.- Unfortunately in this retrospective series of RM, epigenetic alterations could not be thoroughly investigated, preliminary immunohistochemical staining performed for H3K27me3 in a subset of tumors provided heterogeneous and thereby inconclusive results, that we decided not to include in the present manuscript.

Reviewer 2 Report

This study on Rhabdoid Meningiomas (RM) comprehends a literature review and an authors’ series of 23 patients (33 meningioma specimens). The literature’s reported cases evidence the heterogeneity of RM: for example, their prevalence in adults and their presence in childhood, the variety of aggressiveness (WHO grade I through III, with prevalence of grade II-III), the predominance of mixed histological patterns in most cases. The genetic analysis of the authors’ series of RM allowed an interesting subdivision into two classes, one characterized by chromosomal losses and the other by both chromosomal losses and gains: multiple alterations in chromosomal number were associated to poor prognosis.

The paper describes nicely the chapter of RM giving a panoramic view of these rare tumors: clinical, histopathologic, and genetic features are reported, condensing the most important issues on the matter.

Author Response

Comment 1.This study on Rhabdoid Meningiomas (RM) comprehends a literature review and an authors’ series of 23 patients (33 meningioma specimens). The literature’s reported cases evidence the heterogeneity of RM: for example, their prevalence in adults and their presence in childhood, the variety of aggressiveness (WHO grade I through III, with prevalence of grade II-III), the predominance of mixed histological patterns in most cases. The genetic analysis of the authors’ series of RM allowed an interesting subdivision into two classes, one characterized by chromosomal losses and the other by both chromosomal losses and gains: multiple alterations in chromosomal number were associated to poor prognosis.

The paper describes nicely the chapter of RM giving a panoramic view of these rare tumors: clinical, histopathologic, and genetic features are reported, condensing the most important issues on the matter”.

Answer to comment 1.- We appreciate the reviewer's comments on the manuscript and its contents and we have nothing to add in this regard.
